# Hematological changes in the blood of experimental male and female albino rats on exposure to pesticide, dimethoate

Nida Saleem[ID]1*, Mushtaq Hussain Lashari2*, Hafiz Ishfaq Ahmad2, Shazia Tahreem3, Mikhlid H. Almutairi3, Shakeel Ahmed3,4

**1** Department of Zoology, The Islamia University, Bahawalpur, Pakistan, **2** Department of Animal Breeding and Genetics, Faculty of Veterinary and Animal Sciences, The Islamia University, Bahawalpur, Pakistan, **3** Zoology Department, College of Science, King Saud University, Riyadh, Saudi Arabia, **4** Department of Experimental Medicine, University of Campania "Luigi Vanvitelli", Naples, Italy

* mushtaq.hussain@iub.edu.pk (MHL); nidasaleem1117@gmail.com (NS)

## Abstract

### Objective

Evaluating pesticides' impacts on human health, ecological balance, and agricultural production is the main focus of research on pesticides. The present study aims to investigate the hematological parameters of both genders of albino rats after exposure to dimethoate pesticides.

### Material and Methods

There was a total of eight groups of albino rats, there were two control groups, and treatment groups were further divided into six groups. A low dose of: 5ml was given to the T1 group, a medium dose of 10ml was given to the T2 group, and a high dose of 20 ml was given to the T3 group. Each group contained nine rats in them (n=9). The first two groups of albino rats were taken as the control group and placed in normal conditions. The other six groups of albino rats were administered sub-lethal doses of dimethoate pesticide by mixing this pesticide in food by oral gavage for 90 days.

### Results

The study showed that the number of white blood cells, platelets count, granulocyte count and lymphocyte count in both genders of albino rats after 30-, 60- and 90-days exposure to dimethoate was significantly increased where simultaneously the number of red blood cells, hemoglobin level, mean corpuscle hemoglobin count, and mean corpuscle hemoglobin concentration count was decreased considerably due to exposure of dimethoate pesticide in both genders of albino rats. No changes observed in the control group of male and female albino rats. This study concluded that dimethoate pesticide effects the blood parameter of male and female albino rats.

**Data availability statement:** All relevant data are within the paper.

**Funding:** The author(s) received no specific funding for this work.

**Competing interests:** The authors have declared that no competing interests exist.

**Abbreviations:** RBC, Red blood cells; WBC, White blood cells; PLT, Platelets; HGB, Hemoglobin; HCT, Hematocrit; MCH, Mean corpuscular hemoglobin; MCHC, Mean corpuscular hemoglobin concentration; LYM, Lymphocyte; GRA, Granulocyte; MPV, Mean platelet volume; NK Cells, Natural killer cells; CBC, Complete blood count

## Conclusion

The study's findings highlight the importance of strict regulatory policies and thorough risk assessment techniques in order to reduce the harmful impacts of pesticides on the health of people and the environment.

## Introduction

The largest sector of Pakistan economy is agriculture. Most people are either directly or indirectly depending on this industry. It is the main source of foreign exchange profits, makes up half of the employed labor force, and provides roughly 24% of the GDP [1]. In Pakistan the era of pesticide can be divided into six periods. At that time Pakistan largest imports were for malaria control program, locust control program, sugarcane, aerial spray of cotton, rice and tobacco. Pesticides are shipped in large containers and many accidents were occurred during this shipment. In Pakistan almost one-third of agricultural products are produced which depends on pesticide application by [2].

Without pesticides uses, there would be loss in fruit production almost 78%, 54% loss in production of vegetable, or 32% loss in production of cereal. That is why, pesticides play a very important role in reducing diseases and worldwide increase in the yield of crops [3]. A pesticide is a chemical or a biological agent such as a virus, antimicrobial, fungal, bacterium or disinfectant that incapacitates or kill pests. The use of pesticide is very common and we use the term pesticide for plant protection product. It is used to eliminate or control wide variety of pests that can damage agricultural industry, crops and livestock and also reduce the productivity of farm [4].

Pesticide exposure, particularly to organophosphate (OP) chemicals, accounts for a significant number of cases of acute poisoning. Acetylcholinesterase (ache) enzyme inhibition is the main mechanism of action for OP insecticides [5]. Acetylcholine (ach) builds up in the nervous system after pain is inactivated, which over stimulates muscarinic and nicotinic receptors. The three main kinds of poisoning symptoms and signs include muscarinic, nicotinic, and effects on the central nervous system. Concerns over the indiscriminate and excessive use of pesticides, the resulting environmental contamination, and the harmful consequences on human health are increasing. The risks associated with applying these chemicals have recently been brought to light by an abrupt rise in their use in household and government use, as well as in rats.

One of the most important organophosphorus pesticides, i.e., dimethoate, is extensively used for indoor control of houseflies. It is effective against a variety of insects and mites. The main human populations at risk of high levels of dimethoate exposure include farmers, pesticide workers, and pesticide producers [6]. Even though dimethoate is considered to be moderately hazardous (acute oral LD50 310 mg/kg/day, and estimated value for humans 30 mg/kg), it usually blocks neuromuscular transmission in humans as well as animals [7].

An organophosphorus insecticide named as dimethoate has high water solubility. When it is in pure form, it is crystalline solid, white in color and a mercaptan odor. It is stable in aqueous solutions with a pH range of 2–7, although it becomes unstable in alkaline solutions at ambient temperature. It has a low affinity for organic substances and a weak affinity for soils. It is quite resistant to microbial deterioration, susceptible to hydrolysis in acidic environments, and non-volatile as evidenced by its low vapor pressure [8].

There are many studies done on dimethoate's impact on cancer of the gastrointestinal tract. The current work aims to examine the changes in p16, Bcl-2, and c-myc gene expression in mouse digestive tissue after perfusion with the widely used in Chinese agricultural regions organophosphorus pesticides dichlorvos and dimethoate [9]. A decrease in sperm count and infertility in humans have been linked to the overproduction of reactive oxygen species (ROS) by OPs insecticides in both intracellular and extracellular areas. Dimethoate (DMT), one of the OP pesticides, is used in agriculture for protecting crops like apples, pears, and other fruits against aphids and leaf miners [10].

Despite the fact that OP pesticides are known to suppress acetylcholinesterase activity in the central (CNS) and peripheral nervous systems [11] a number of observations point to the possibility that cholinergic system disruptions are not the only cause of OP neurotoxicity, particularly in the case of chronic exposure [12].

The main objective of this study is to investigate the effects of dimethoate pesticide on the hematological parameters of male and female albino rats by using the method of CBC. Blood parameters are important for diagnosing the functional properties of organisms exposed to pollutants.

## Materials and methods

The present experiment was conducted at Department of Zoology, The Islamia University Bahawalpur. Healthy albino rats (n = 72) of both sexes (male (n = 36) and female (n = 36) weighing 160 g were purchased from the local market, each group consisting of nine rats. Two groups were used as the control, and the other six groups orally received different doses of dimethoate pesticide were shown in Table 1. The pesticide was obtained in a pure form from reputable pesticide market (Bayer pharmaceutical company) and different concentrations were prepared. Three concentrations were prepared according to the average body weight.

### Toxicity test and preparing dosages for exposure

There was total eight groups, four groups of females, and four groups of male albino rats. There were two control groups and treatment groups were further divided into six groups (T1F, T1M, T2F, T2M, T3F, T3M) which were exposed to

**Table 1. Experimental groups.**

| Treatment Groups | Male (4 Groups) | Female (4 Groups) |
|---|---|---|
| 1.Control group | 9 albino rats (CM1, CM2, CM3, CM4, CM5, CM6, CM7, CM8, CM9) | 9 albino rats (CF1, CF2, CF3, CF4, CF5, CF6, CF7, CF8, CF9) |
| 2.Treated group 1 (5ml/80-160g) | 9 albino rats (T1M1, T1M2, T1M3, T1M4, T1M5, T1M6, T1M7, T1M8, T1M9) | 9 albino rats (T1F1, T1F2, T1F3, T1F4, T1F5, T1F6, T1F7, T1F8, T1F9) |
| 3.Treated group 2 (10ml/80-160g) | 9 albino rats (T2M1, T2M2, T2M3, T2M4, T2M5, T2M6, T2M7, T2M8, T2M9) | 9 albino rats (T2F1, T2F2, T2F3, T2F4, T2F5, T2F6, T2F7, T2F8, T2F9) |
| 4.Treated group 3 (20ml/80-160g) | 9 albino rats (T3M1, T3M2, T3M3, T3M4, T3M5, T3M6, T3M7, T3M8, T3M9) | 9 albino rats (T3F1, T3F2, T3F3, T3F4, T3F5, T3F6, T3F7, T3F8, T3F9) |

different concentrations of dimethoate pesticides. Low dose: 5ml/kg was given to T1 group, medium dose: 10ml/kg was given to T2 group and high dose: 20 ml/kg was given to T3 group. Different concentrations of pesticides were prepared according to calculated $LD_{50}$ (acute oral $LD_{50}$ 0.085 ml). T1 group was treated with low concentration of pesticide ($\frac{1}{40}$ of $LD_{50}$), T2 group was treated with medium concentration of pesticide ($\frac{1}{20}$ of $LD_{50}$), T3 group was treated with high concentration of pesticide ($\frac{1}{10}$ of $LD_{50}$). Each group contained nine rats in them (n = 9).

### Designing the experiment and acclimatization of albino rats

Albino rats of both sexes, approximately 20 week's age, was purchased from local market. All rats were examined for disease condition and acclimatized to the laboratory environment about 20 days before use and maintained under controlled laboratory conditions, including 12 h dark/light cycle, standard temperature or relative humidity. Animals were kept in separate cage with height of no less than 5 inches (12.7 cm) and a floor area per mouse of 6–15 inches (38.7 to 96.7 cm) squared, depending on body weight, and given a standard diet (laboratory chow) or water ad libitum during lab trial. The animals in the treated groups were given dimethoate one time in a week for 90 days in a row, while the animals in the control group received their daily diet of common laboratory chow. Every week during the trial, all animals were weighed. No mortality was recorded at different days and blood was collected by puncturing heart to perform hematological tests.

### Sample collection

A total of 24 blood samples (three rats from each male and female group) from albino rats were taken after every 30 days in 90 days of trial with 5ml auto disposable plastic syringes, with plastic needle, needle size 23 G × 1 inserted into ventral side of heart keeping in view the recommended protocols of rat euthanasia, and blood was collected [13]. Isoflurane inhalation anesthesia was used to anesthetize experimental rats for blood collection in hematology studies due to its rapid onset and quick recovery.

Three rats from eight groups were dissected and blood was collected. Syringes were washed with prepared EDTA solution to prevent clotting of blood. Syringe was inserted into the heart side to puncture heart then instantly blood was entered into syringe. Collected blood was put into the EDTA vial to prevent clotting. Maximum blood sample was taken from each albino rat.

The blood samples were analyzed for hematological parameters. The following hematological parameters were studied: Red Blood Cells (RBCs) and White Blood Cells (WBCs), hemoglobin (Hb), Packed Cell Volume (PCV), MCH, MCHC, GRA (Granulocyte), LYM (Lymphocyte) and Platelets (PLT) count. The results are shown in Fig 1.

### Ethical consideration

All research work in the submitted paper has been conducted ethically and responsibly, and is in full compliance with all relevant codes of experimentation and legislation. This study is approved by the ethical committee of The Islamia University of Bahawalpur. The meeting was held on 22-04-2023 with approval number 9357-IUB. This study was reported in accordance with ARRIVE guidelines.

### Statistical analysis

Level of significance at exposure times and pesticide concentrations for the data collected from the control and experimental groups was undertaken using IBM SPSS statistic 20, One-way ANOVA was applied. Both the mean and the standard deviation were determined for numerical variables such as hematological parameters (hemoglobin (HGB), red blood cell (RBC), white blood cell (WBC), granulocyte (GRA), lymphocyte (LYM), mean corpuscle hemoglobin (MCH), mean corpuscle hemoglobin concentration (MCHC), platelet (PLT).

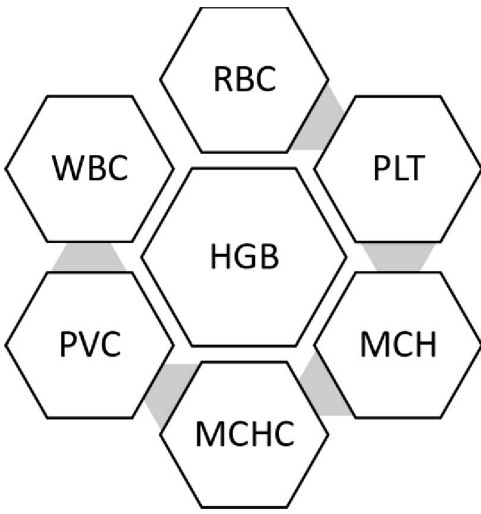

**Fig 1. Hematological parameters.** Hematological analyses were performed through manual blood counting [14] and cross-checked using the Rayto RT-7600 Vet Auto Hematology Analyzer (China).

## Results and discussion

### Blood parameters

Hematological results of albino rats of both genders after 30-, 60- and 90-days exposure of dimethoate pesticide were shown in Tables. Albino rats expose to dimethoate pesticide for 90 days induces more changes than the rats exposed to 60 or 30 days.

In comparison to a control group, the Table 2. shows the effects of different dosages of dimethoate pesticide (5 ml, 10 ml, and 20 ml) on a number of hematological changes in male and female albino rats over the course of a 30-day period. The statistical significance of each parameter is shown by the p-value, and it is represented as mean ± SEM (Standard Error of the Mean). The effects of dimethoate on hematological parameters are shown in Fig 2.

When compared with control albino rats, there were significant decrease in hemoglobin count P ≤ 0.00 (♀) P ≤ 0.02 (♂), mean corpuscle hemoglobin P ≤ 0.001 (♀) P ≤ 0.001 (♂), and hematocrit level P ≤ 0.04 (♂) of male and female albino rats treated with dimethoate. Generally, there was also a significant increase in the white blood cells P ≤ 0.02 (♀) P ≤ 0.05 (♂), mean corpuscular hemoglobin concentration P ≤ 0.03 (♀) P ≤ 0.02 (♂) Platelets count P ≤ 0.01 (♀) P ≤ 0.01 (♂) lymphocyte P ≤ 0.01 (♀) P ≤ 0.003 (♂) granulocyte count P ≤ 0.02 (♀) P ≤ 0.01 (♂ and hematocrit level P ≤ 0.02 (♀). Red blood cells significantly decrease in female P ≤ 0.01, while significantly increase in male group of albino rats P ≤ 0.02.

In comparison to a control group, the Table 3. shows the effects of different dosages of dimethoate pesticide (5 ml, 10 ml, and 20 ml) on a number of hematological changes in male and female albino rats over the course of a 30-day period. The statistical significance of each parameter is shown by the p-value, and it is represented as mean ± SEM (Standard Error of the Mean). The effects of dimethoate on hematological parameters are shown in Fig 3.

When compared with control albino rats, there were significant decrease in red blood cells count P ≤ 0.02 (♀) P ≤ 0.003 (♂), hemoglobin count P ≤ 0.02 (♀) P ≤ 0.01 (♂), lymphocyte P ≤ 0.00 (♀) P ≤ 0.003 (♂) and hematocrit level P ≤ 0.04 (♂) of male and female albino rats treated with dimethoate. Generally, there was also a significant increase in the white blood cells P ≤ 0.00 (♀) P ≤ 0.009 (♂), mean corpuscular hemoglobin concentration P ≤ 0.004 (♀) P ≤ 0.001 (♂) granulocyte count P ≤ 0.00 (♀) P ≤ 0.001 (♂), (♀) and platelets count P ≤ 0.00 (♀) P ≤ 0.009 (♂) and non-significant decrease in hematocrit level P ≤ 0.4. There is significant increase in mean corpuscle hemoglobin P ≤ 0.02 (♀) and significant decrease in mean corpuscle hemoglobin P ≤ 0.02 (♂).

**Table 2. Alterations in CBC parameters in albino rats of both genders after 30 days of administration of dimethoate presented in Mean ± S.E.M. and their significance level.**

| Parameters | Gender | Control group (n = 3) Mean ± SEM | Treated group 1 5ml (n = 3) Mean ± SEM | Treated group 2 10 ml (n = 3) Mean ± SEM | Treated group 3 20ml (n = 3) Mean ± SEM | P Value |
|---|---|---|---|---|---|---|
| WBC ($10^3/\mu$L) | F (♀) | 3.06 ± 0.3 | 3.74 ± 0.2 | 4.95 ± 0.5 | 6.22 ± 0.6 | .002 |
| | M (♂) | 3.68 ± 0.4 | 5.36 ± 0.5 | 5.47 ± 0.4 | 5.82 ± 0.7 | 0.05 |
| RBC ($10^6/\mu$L) | F (♀) | 8.48 ± 0.1 | 7.44 ± 0.1 | 7.25 ± 0.1 | 7.01 ± 0.3 | .001 |
| | M (♂) | 7.59 ± 0.2 | 8.70 ± 0.3 | 8.21 ± 0.1 | 8.17 ± 0.2 | 0.02 |
| HGB (g/dL) | F (♀) | 14.50 ± 0.4 | 12.47 ± 0.2 | 11.13 ± 0.3 | 11.10 ± 0.2 | .000 |
| | M (♂) | 11.97 ± 0.3 | 12.57 ± 0.2 | 11.30 ± 0.5 | 10.57 ± 03 | 0.02 |
| PLT ($10^3/\mu$L) | F (♀) | 121.7 ± 1.3 | 97.67 ± 0.8 | 85.00 ± 3.6 | 544.33 ± 1.4 | 0.01 |
| | M (♂) | 158.3 ± 17 | 215 ± 41 | 79.67 ± 4.7 | 387.67 ± 15 | 0.1 |
| HCT (%) | F (♀) | 44.83 ± 1.8 | 45.67 ± 0.2 | 39.2 ± 0.2 | 41.13 ± 1.7 | .023 |
| | M (♂) | 47.73 ± 1.8 | 37.00 ± 1.7 | 38.7 ± 1.2 | 37.53 ± 3.8 | 0.04 |
| MCH (pg) | F (♀) | 17.33 ± 1.4 | 14.03 ± 0.4 | 13.20 ± 0.5 | 11.43 ± 0.6 | 0.01 |
| | M (♂) | 17.20 ± 1.0 | 15.33 ± 0.3 | 14.20 ± 0.3 | 13.37 ± .0.6 | 0.01 |
| MCHC (g/dL) | F (♀) | 27.20 ± 0.3 | 27.57 ± 0.1 | 26.33 ± 1.1 | 30.37 ± 0.9 | 0.03 |
| | M (♂) | 26.77 ± 0.3 | 28.30 ± 0.1 | 27.20 ± 0.3 | 27.90 ± .0.2 | 0.02 |
| LYM ($10^3/\mu$L) | F (♀) | 2.69 ± 0.0 | 2.91 ± 0.3 | 4.28 ± 0.5 | 4.41 ± 0.2 | 0.01 |
| | M (♂) | 2.65 ± 0.1 | 2.69 ± 0.2 | 3.41 ± 0.3 | 4.28 ± .0.4 | .003 |
| GRA ($10^3/\mu$L) | F (♀) | 0.79 ± 0.1 | .58 ± 0.1 | .97 ± 0.2 | 1.59 ± 0.2 | .021 |
| | M (♂) | .57 ± 0.1 | .24 ± 0.2 | .25 ± 0.1 | .28 ± 0.1 | 0.01 |

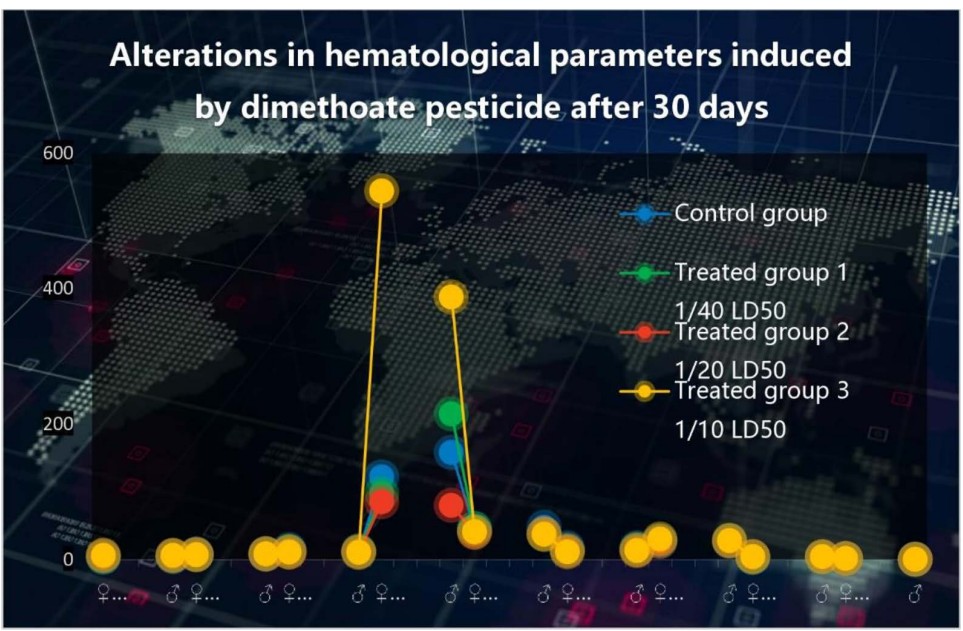

**Fig 2. Alterations in hematological parameters induced by dimethoate after 30 days.** The figure represents the changes in hematological parameters among different experimental groups. The blue dots indicate the control group, while the green, red and yellow markers represent treated group exposed to different sublethal doses of dimethoate pesticide: Treated Group 1 (1/40 LD50), Treated Group 2 (1/20 LD50), and Treated Group 3 (1/10 LD50). Data points depict individual subjects, with the vertical axis representing hematological values. The lines connecting points indicate the trends observed in different groups.

**Table 3. Alterations in CBC parameters in albino rats of both genders after 60 days of administration of dimethoate presented in Mean±S.E.M. and their significance level.**

| Parameters | Gender | Control group (n=3) Mean±SEM | Treated group 1 5ml (n=3) Mean±SEM | Treated group 2 10ml (n=3) Mean±SEM | Treated group 3 20ml (n=3) Mean±SEM | P Value |
|---|---|---|---|---|---|---|
| WBC ($10^3$/µL) | F (♀) | 3.74±0.1 | 4.35±0.1 | 5.49±0.1 | 6.50±0.1 | .000 |
|  | M (♂) | 4.59±0.3 | 3.08±0.4 | 4.28±0.3 | 2.95±0.3 | .009 |
| RBC ($10^6$/µL) | F (♀) | 9.40±0.3 | 7.73±0.7 | 6.91±0.4 | 6.26±0.7 | 0.02 |
|  | M (♂) | 9.49±0.3 | 7.07±0.4 | 7.20±1.0 | 4.57±0.4 | .003 |
| HGB (g/dL) | F (♀) | 22.33±0.3 | 17.93±1.7 | 16.77±1.4 | 14.67±1.5 | 0.02 |
|  | M (♂) | 21.80±0.7 | 20.53±0.8 | 19.03±1.0 | 13.33±2.4 | 0.01 |
| PLT ($10^3$/µL) | F (♀) | 155.0±15 | 463.0±11 | 158.33±35 | 278.67±52 | 0.03 |
|  | M (♂) | 127.33±4.1 | 475.33±41 | 664.0±59 | 813.67±67 | .000 |
| HCT (%) | F (♀) | 44.40±1.3 | 44.97±4.5 | 45.97±4.3 | 35.83±5.8 | 0.4 |
|  | M (♂) | 45.10±3.5 | 42.70±1.3 | 49.83±2.3 | 55.63±2.8 | 0.03 |
| MCH (pg) | F (♀) | 27.00±0.8 | 15.43±0.7 | 19.17±3.4 | 16.53±2.0 | 0.02 |
|  | M (♂) | 21.77±1.6 | 25.67±0.8 | 26.40±0.5 | 27.53±0.9 | 0.02 |
| MCHC (g/dL) | F (♀) | 25.83±1.2 | 41.27±4.3 | 42.83±3.0 | 45.70±0.6 | .004 |
|  | M (♂) | 23.87±0.9 | 50.87±0.9 | 47.70±0.3 | 36.23±5.7 | .001 |
| LYM ($10^3$/µL) | F (♀) | 2.84±0.1 | 2.04±0.2 | 1.28±0.2 | 1.36±0.1 | .000 |
|  | M (♂) | 2.61±0.3 | 2.28±0.7 | 1.81±0.1 | 1.31±0.1 | .003 |
| GRA ($10^3$/µL) | F (♀) | 0.98±0.1 | 2.31±0.1 | 2.59±.0.1 | 2.84±0.2 | .000 |
|  | M (♂) | 2.31±.0.2 | 0.85±0.3 | 2.52±0.2 | 2.65±0.2 | .001 |

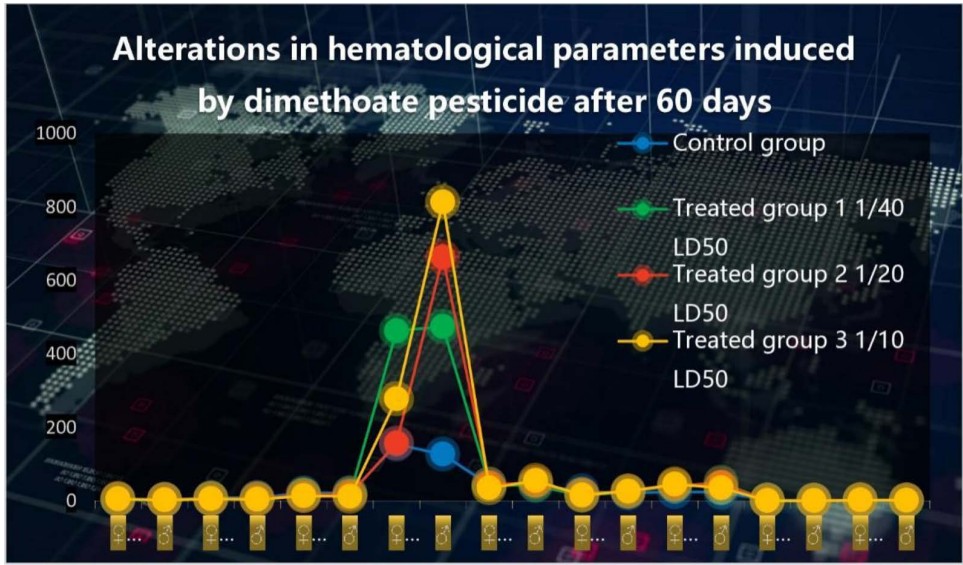

**Fig 3. Alterations in hematological parameters induced by dimethoate after 60 days.** The figure illustrates the changes in hematological parameters among different experimental groups after 60 days of dimethoate exposure. The blue dots represent the control group, while the green, red, and yellow markers denote treated groups exposed to varying sub-lethal doses of dimethoate: Treated Group 1 (1/40 LD50), Treated Group 2 (1/20 LD50), and Treated Group 3 (1/10 LD50). The vertical axis represents hematological parameter values, and the data points reflect individual subjects. The observed peak in the treated groups indicates significant alterations in response to pesticide exposure.

In comparison to a control group, the Table 4. shows the effects of different dosages of dimethoate pesticide (5 ml, 10 ml, and 20 ml) on a number of hematological changes in male and female albino rats over the course of a 30-day period. The statistical significance of each parameter is shown by the p-value, and it is represented as mean ± SEM (Standard Error of the Mean). The effects of dimethoate on hematological parameters are shown in Fig 4.

When compared with control albino rats, there were significant decrease in lymphocyte $P \leq 0.004$ (♀) $P \leq 0.003$ (♂) and hematocrit level $P \leq 0.04$ (♂) of male and female albino rats treated with dimethoate. Generally, there was also a significant increase in the red blood cells count $P \leq 0.04$ (♀) $P \leq 0.001$ (♂), white blood cells $P \leq 0.002$ (♀) $P \leq 0.003$ (♂), mean corpuscular hemoglobin concentration $P \leq 0.02$ (♀) $P \leq 0.01$ (♂) hematocrit level $P \leq 0.02$ (♀) $P \leq 0.001$ (♂) mean corpuscle hemoglobin $P \leq 0.04$ (♀) $P \leq 0.01$ (♂) granulocyte count $P \leq 0.03$ (♀) $P \leq 0.04$ (♂), and platelets count $P \leq 0.05$ (♀) $P \leq 0.003$ (♂). There is a significant increase in hemoglobin count $P \leq 0.00$ (♀) and decrease in $P \leq 0.00$ (♂).

## Discussion

Albino rats exposed to dimethoate for 60 and 90 days exhibit more changes in the blood composition in both genders of albino rats than those treated for 30 days. Red blood cells, hemoglobin, and hematocrit values were significantly decreased in albino rats exposed to dimethoate pesticide. The percentage of inhibition increased with the duration of exposure. White blood cell, granulocyte, and lymphocyte values were significantly increased in albino rats exposed to dimethoate pesticide.

The CBC parameters have been used in the diagnosis of inflammation and infections during disease conditions [15]. Hematological parameters are influenced by hormones [18]. In the present study, hematological parameters in the levels of hemoglobin, red blood cells, or other parameters of blood such as packet cell volume, mean corpuscular hemoglobin,

**Table 4. Alterations in CBC parameters in albino rats of both genders after 90 days of administration of dimethoate presented in Mean±S.E.M. and their significance level.**

| Parameters | Gender | Control group (n=3) Mean±SEM | Treated group 1 5ml (n=3) Mean±SEM | Treated group 2 10ml (n=3) Mean±SEM | Treated group 3 20ml (n=3) Mean±SEM | P Value |
|---|---|---|---|---|---|---|
| WBC ($10^3$/µL) | F (♀) | 3.98±0.5 | 3.99±0.3 | 5.39±0.4 | 6.53±0.3 | .002 |
| | M (♂) | 3.75±0.3 | 7.23±1.4 | 7.85±0.3 | 10.74±0.9 | .003 |
| RBC ($10^6$/µL) | F (♀) | 6.45±0.4 | 6.38±0.1 | 6.85±0.1 | 8.00±0.5 | 0.04 |
| | M (♂) | 6.81±0.3 | 8.79±0.2 | 7.91±0.3 | 6.60±0.3 | .001 |
| HGB (g/dL) | F (♀) | 13.10±0.6 | 20.13±0.9 | 24.00±0.1 | 21.40±0.5 | .000 |
| | M (♂) | 14.90±0.1 | 12.63±0.5 | 12.27±0.4 | 10.67±0.4 | .000 |
| PLT ($10^3$/µL) | F (♀) | 128.00±7.5 | 463.33±17 | 618.00±36 | 511.25±79 | 0.05 |
| | M (♂) | 186.00±52 | 570.33±57 | 713.00±10 | 781.00±82 | .003 |
| HCT (%) | F (♀) | 34.00±0.6 | 31.47±0.6 | 36.05±1.2 | 42.00±2.9 | 0.02 |
| | M (♂) | 38.90±1.0 | 42.10±0.6 | 44.83±1.5 | 47.60±0.4 | .001 |
| MCH (pg) | F (♀) | 21.62±1.1 | 32.13±1.3 | 35.25±0.3 | 22.18±4.2 | 0.04 |
| | M (♂) | 21.87±0.3 | 31.67±1.3 | 29.30±2.0 | 28.27±1.9 | 0.01 |
| MCHC (g/dL) | F (♀) | 33.20±1.9 | 63.73±1.6 | 64.05±0.9 | 33.10±9.9 | 0.02 |
| | M (♂) | 33.20±0.7 | 24.57±0.1 | 42.07±6.6 | 47.37±3.9 | 0.01 |
| LYM ($10^3$/µL) | F (♀) | 4.07±0.6 | 2.04±0.3 | 1.26±0.1 | 1.73±0.4 | .004 |
| | M (♂) | 4.44±0.4 | 3.57±0.1 | 3.84±0.4 | 2.13±0.1 | .003 |
| GRA ($10^3$/µL) | F (♀) | .93±0.2 | 3.69±0.7 | 3.92±0.6 | 4.17±0.9 | 0.03 |
| | M (♂) | 2.27±0.6 | 3.68±0.2 | 2.53±0.2 | 2.16±0.1 | 0.04 |

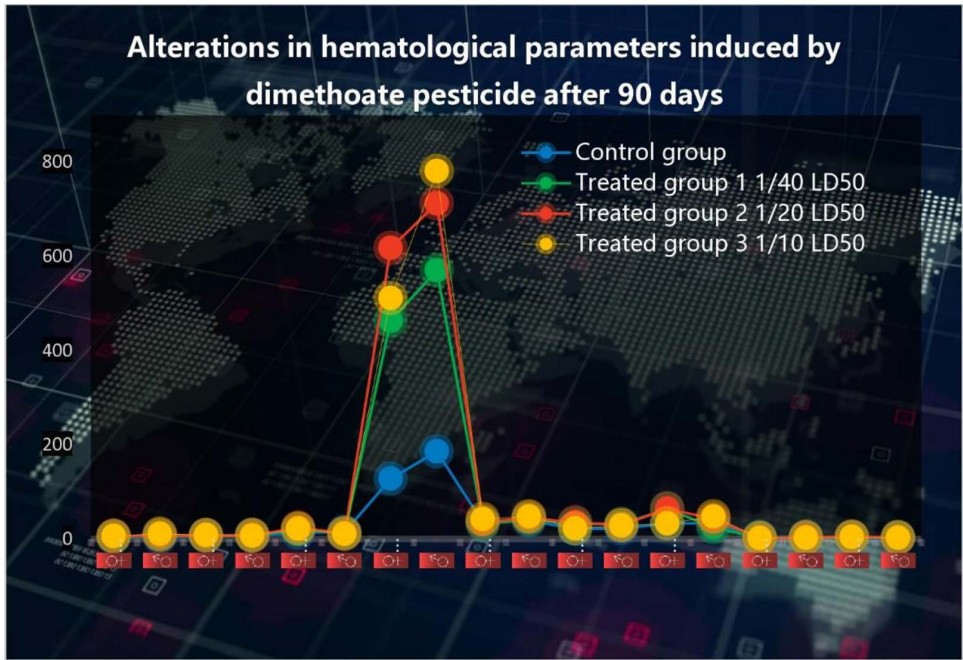

**Fig 4. Alterations in hematological parameters induced by dimethoate after 90 days.** The figure represents changes in hematological parameters among different experimental groups. The blue dots indicate the control group, while the green, red, and yellow markers represent treated groups exposed to varying sub-lethal doses of dimethoate: Treated Group 1 (1/40 LD50), Treated Group 2 (1/20 LD50), and Treated Group 3 (1/10 LD50). The vertical axis denotes hematological parameter values, while the data points and lines illustrate trends in different treatment groups over time.

or mean corpuscular hemoglobin concentration dose-dependent decrease were observed in all treated groups. These results are in agreement with [16].

In the present study, levels of hemoglobin and red blood cell count were decreased in both genders of albino rats. As the number of doses increases, the number of red blood cell counts also decreases and the result are in agreement with the result of [17] in which the number of red blood cells also decreased. When complement activates hemolysin, an immunogenic new toxin with hemolytic properties, red blood cells are damaged, and the antibody is dissolved. This can set off a series of events that include anemia [19].

According to the present study, dimethoate induced changes in experimental albino rats in terms of hematological indices. In present studies, hemoglobin, hematocrit, and red blood cell count were significantly decreased, which indicates the presence of anemia induced by dimethoate pesticide. These results are in agreement with [20]. The hematological changes were mainly observed in the hemoglobin level and hematocrit level, which is due to malabsorption of nutrients or hyperactivity of animal [21]. Our findings are in agreement with [21], who showed the decrease in the hemoglobin level and the decrease in the red blood cells is due to the effect of pesticides on the organs that form blood, including bone marrow and liver, or inhibition may be due to biosynthesis of heme.

Pesticide residues that cause poisoning lead to anemia development due to the interference of biosynthesis of hemoglobin and shorten the erythrocyte's life span [20,22]. Our finding is in agreement with the [22], which showed a decrease in red blood cell count and hemoglobin levels due to dimethoate pesticide, [17], who noticed hemoglobin and red blood cell reduction or increase in sedimentation rate of erythrocytes in albino rats, exposure to oral doses of three different concentrations of dimethoate pesticide. Because dimethoate may interfere with red blood cells' ability to transport oxygen, exposure to it in albino rats may result in a drop in hemoglobin levels. Because dimethoate inhibits cholinesterase action,

erythropoiesis may be disrupted, resulting in a drop in red blood cell synthesis and a simultaneous fall in hemoglobin levels. Furthermore, dimethoate-induced oxidative stress and red blood cell destruction may be involved in the albino rats' post-exposure decrease in hemoglobin levels.

The effects of organophosphorus pesticides may be due to their ability to produce free radicals [5,23]. The results of some previous studies on methidathion show that use of antioxidants, vitamins A, C, or E, overcome methidathion toxicity [17]. This is the fact that ensure this hypothesis of the ability of the organophosphorus pesticide to produce free radicals, which will implicate like playing a role in etiology of some hematological alterations [24].

In the present study, the level of white blood cell count was increased in both genders of albino rats. As the number of doses increases, the number of white blood cell counts also increases and result are in agreement with the result of [25]. When albino rats are exposed to pesticides, their white blood cell (WBC) count may rise in reaction to inflammation or immune system activation brought on by the pesticide's harmful effects. This increase in WBC count may be a sign that the body is trying to protect itself against the negative consequences of the pesticide, such as oxidative stress or tissue damage. Furthermore, as part of the body's defensive response, certain pesticides may directly cause the bone marrow to create more white blood cells. Assessing the toxicity levels of the pesticide and the physiological response of albino rats to pesticide exposure can both be aided by tracking the white blood cell count (WBC). A statistically significant increase was detected in white blood cells (lymphocyte and granulocyte) in the treated group compared to the non-treated group (P<0.05).

[26] showed an increase in white blood cell counts in acute poisoning of organophosphate pesticide. [27], stated higher white blood cell counts (monocytes and lymphocytes) in farmers. White blood cell counts or platelet counts increased significantly in the exposed group compared with the non-exposed group. The prevalence of thrombocytosis significantly increased in the exposed group compared with the non-exposed group. Previous studies assessed exposure to pesticides or many hematopoietic cancers. Meta-analysis (a review of 14 studies published in between 1984 or 2004) shows that pesticide occupational exposure increases the risk of leukemia in adults by [28].

Platelets are the coagulating agents for the incidence of wounds. Greater concentrations of platelets may lead to coagulation of blood in the vessels, whereas deficiency causes a problem in wound healing. Dimethoate exposure in albino rats cause an increase in platelet count because the pesticide causes a pro-inflammatory response or interfere with normal hematopoiesis. As the body reacts to the toxic insult, exposure to dimethoate cause the release of cytokines or other mediators that promote platelet synthesis in the bone marrow, increasing the platelet count. Monitoring platelet counts can help assess the possible negative effects of dimethoate on coagulation and hemostasis in albino rats, as well as provide insights into the hematological consequences of exposure to the drug [29].

Higher platelets are found in females with higher estradiol levels [30]. Platelet count was significantly increased in the treated group in present study, which was in agreement with the results of previous studies. Abu Mourad reported a significant increase in the mean platelet counts in farmers as compared to control group (P<0.05). The prevalence of thrombocytosis in the exposed group of farmers was 9.09% in previous study and production of platelet increases due to inflammation or high oxidative stress. [27], also showed that long-term exposure to pesticides was accomplished by oxidative stress. Thrombocytosis prevalence was increases in the participants with more work experience in our study, which could indirectly suggest the level of exposure to pesticides. Occupational exposure started at all stages of pesticide formulation, which include manufacture or application which involve exposure to mixtures of different types of chemicals. Susceptibility of Individual to pesticides is very important in toxicity of pesticide. Our findings have similar results with [31], which reported increase in platelet counts in pesticide retailer rather than in the controls. In contrary, [32] reported that there is no difference in platelet count between farm workers exposed to pesticide or control subjects. To the best-known study, there is present only one study about pesticide exposure which effect mean platelet volume in the farm workers. In this study, [32] reported that the decrease in mean platelet volume is present in 15% sprayers of greenhouse exposed to pesticides. Our findings confirm the result of this study.

In present study the level of lymphocyte count was increased in both genders of albino rats. As the number of doses increases, the number of lymphocyte count were also increased and result are in agreement with the result of [33]. In an effort to neutralize the toxic effect, dimethoate can cause immune system disruption, which will cause lymphocytes to proliferate. The body's attempt to lessen the harm caused by the pesticide and restore immunological homeostasis in albino rats is shown in this elevated lymphocyte count. The body uses lymphocytes as its defense against outside invaders. These may include natural killer cells, T cells, and B cells. Lymphocyte count was significantly decreased in treated group in our study, which was in the agreement with the results of previous studies control. These results were in accordance with [34]. The eventual increase in neutrophils or decrease in lymphocyte count due to inflammation, which eventually increases baseline neutrophil-lymphocyte ratio. Chronic exposure to dimethoate pesticide likely to cause change in the hematological parameters in normal range in this study.

In our study, neutrophil, lymphocyte or platelet ratios were significantly higher in the more severely treated rats. Basophils and eosinophils defend the body against parasites, allergens, and other pathogens [35]. Previous studies have reported that the ratio of neutrophil-lymphocyte is a strong indicator of inflammation or prognosis in many diseases. In many experimental studies, it has been clear that the ratio of neutrophil-lymphocyte increases in pesticide-poisoning groups [36].

A result of study suggested that organophosphorus or carbamate pesticides affect the formation of megakaryocytes by cholinergic receptors, which reduce the mean corpuscular hemoglobin count. On the other hand, there is no exact effect of cholinergic substances on the formation of megakaryocytes nor the receptors that mediate these effects, which are identified [37]. The findings of mean corpuscular hemoglobin values are in agreement with those of [37]. Similarly, in another study, it was reported that in patients who have acute poisoning of pesticides, hemoperfusion causes impairment of the platelet aggregation with the incomplete activation of platelet, which associates with the decrease in the formation of thrombin or the increase of fibrinolysis [38], found that there is no difference in the hematological parameters in exposed or non-exposed groups but for white blood cell count which increased in exposed group given pesticides. The difference may be due to the difference in types of pesticides or durations of exposure [28].

Mean corpuscle hemoglobin is the mean of hemoglobin present in the blood. Mean corpuscular hemoglobin concentration is the indication for detecting the hemoglobin amount [35]. In the present study, we found that more treated poisoned rats had increased mean corpuscular hemoglobin and mean corpuscular hemoglobin concentration levels, or this increase was statistically significant. Whereas some experimental studies have reported increased mean corpuscular hemoglobin levels in sub-acute or chronic poisoning of pesticides, others have reported findings of thrombocytopenia [39]. It would be reasonable to conclude that the appearance of thrombocytopenia after oxidative stress negatively affects the membrane of platelet cells or in all blood cells [40]. Many other prospective or controlled studies should confirm these results. A rise in MCH and MCHC may be a sign of hemoglobin content alterations or erythrocyte enlargement brought on by dimethoate-induced oxidative stress or cellular damage. On the other hand, a drop in MCH and MCHC may indicate hemoglobin degradation or compromised erythropoiesis, indicating the pesticide's harmful effects on the synthesis or function of red blood cells in albino rats' bone marrow.

Our findings are consistent with previous studies on organophosphate toxicity, notably in terms of metabolic abnormalities (Author, Year). Elevated AST and ALT levels in exposed rats indicate hepatic distress, which has an indirect impact on erythropoiesis and immunological function. Furthermore, low urea and uric acid levels may indicate poor renal clearance, affecting hematological balance [41].

## Conclusions

In conclusion, different combinations of dimethoate pesticides significantly affected both genders of albino rats' hematology. Exposure to dimethoate causes changes in hematocrit, red blood cell count, hemoglobin levels, white blood cell count, and platelet count, among other blood parameters. Different concentrations of dimethoate pesticides were made.

Statistical differences were seen in the complete blood count parameters ($p < 0.05$). So, it is concluded that these changes could indicate physiological problems and possible health risks related to pesticide use. It has important consequences for Pakistani public health and environmental conservation initiatives that go beyond the confines of the laboratory. Pakistan, a nation that depends largely on agricultural methods, has a difficult time controlling the use of pesticides and the health dangers they cause. The study's findings highlight that having strict rules and careful checks is very important to reduce the harmful effects of pesticides on people and the environment.

## Acknowledgments

The authors extend their appreciation to the Researchers Supporting Project number (RSP2025R191), King Saud University, Riyadh, Saudia Arabia.

## Author contributions

**Conceptualization:** nida saleem.

**Data curation:** nida saleem, Mikhlid H. Almutairi.

**Formal analysis:** nida saleem, Shakeel Ahmed.

**Funding acquisition:** nida saleem, Mikhlid H. Almutairi, Shakeel Ahmed.

**Investigation:** nida saleem.

**Methodology:** nida saleem.

**Project administration:** nida saleem, Mikhlid H. Almutairi, Shakeel Ahmed.

**Resources:** nida saleem, Shazia Tahreem.

**Software:** nida saleem.

**Supervision:** Mushtaq Hussain Lashari, HAFIZ ishfaq AHMAD.

**Validation:** Mushtaq Hussain Lashari, HAFIZ ishfaq AHMAD, Shazia Tahreem.

**Visualization:** nida saleem.

**Writing – original draft:** nida saleem.

**Writing – review & editing:** nida saleem.

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
