## [Decision Letter · Decision Letter 0]

2 Jan 2025

PONE-D-24-37808Hematological changes in the blood of experimental male and female Rattusnorvigacus (Albino Rats) on exposure to pesticide, dimethoatePLOS ONE

Dear Dr. saleem,

Thank you for submitting your manuscript to PLOS ONE. After careful consideration, we feel that it has merit but does not fully meet PLOS ONE’s publication criteria as it currently stands. Therefore, we invite you to submit a revised version of the manuscript that addresses the points raised during the review process.

We look forward to receiving your revised manuscript.

Kind regards,

Ashish Kumar Singh

Academic Editor

PLOS ONE

**Journal Requirements:**

5. Please include your tables as part of your main manuscript and remove the individual files. Please note that supplementary tables (should remain/ be uploaded) as separate "supporting information" files.

**Additional Editor Comments:**

Dear Author's,

Thank you for submitting your manuscript titled "Hematological changes in the blood of experimental male and female Rattus norvegicus (Albino Rats) on exposure to pesticide, dimethoate" (Manuscript ID: PONE-D-24-37808) to PLOS One

The manuscript has been reviewed by two referees, and their evaluations have been carefully considered. Below is a summary of their feedback:

Reviewer 1 has suggested Rejection

Reviewer 2 has recommended minor revisions

Based on these reviews, we are providing you the opportunity to revise your manuscript. To proceed, we request that you:

Address all comments and concerns raised by both reviewers.

Provide a detailed response letter outlining how you have addressed each comment. This will assist the reviewers and editors in evaluating your revisions.

Please note that the revised manuscript will undergo further review to ensure that the revisions adequately address the reviewers’ concerns.

To assist in preparing your revision, I am attaching the full reviewer comments for your reference. We encourage you to carefully consider these and provide clear explanations for any changes made or not made.

Thank you for your effort and commitment to the improvement of your work. We look forward to receiving your revised manuscript.

Reviewer 1

Concern over statistical analysis

authors have not made all data underlying the findings in their manuscript fully available

Study was performed to check effect of pesticide, dimethoate exposure on albino (Albino Rats) hematological parameters.

Study design has some defects, as it is not clear how much dose was administered to rats. Dosage mentioned in mL has very little understanding. Many of the references mentioned in the discussion do mention the dose in form per kg animals.

Lot of studies are mentioned in the discussion tells the effect of dimethoate pesticide on hematological parameters. There is no novelty in the study as MT Akay et al in 1999 mentioned similar observations (PMID: 10509431) All the speculations based on data from semi auto hematology analyzer. If these observations would have been supported with certain biochemical tests it would have great support to the data.

Research group has used both male and female rats for the treatment. However, in results it has not shown comparison of responses from male and female rats. Results are analyzed in very narrow spectrum of treated Vs non-treated.

There is lot of mismatches in description, table and figure

Reviewer 2: Minor revision

Nida Saleem et al. have shown in this article that, following exposure to the dimethoate pesticide for 30, 60, and 90 days in both male and female albino rats, the number of white blood cells, platelet count, granulocyte count, and lymphocyte count significantly increased. In contrast, the number of red blood cells, hemoglobin level, mean corpuscular hemoglobin (MCH), and mean corpuscular hemoglobin concentration (MCHC) were notably decreased. No changes were observed in the control group of male and female albino rats. This study concludes that dimethoate pesticide affects blood parameters in both male and female albino rats.

The article looks interesting to me. I have a few minor comments that could be incorporated to improve it.

1. Since Figure 3.1 appears before Figure 2, please correct the figure numbering so that the figures appear in proper sequence.

2. In Figures 3.1, 2, and 3 (all bar diagrams), label the Y-axis clearly to indicate what is being measured.

3. In the graphical abstract, please include the name of the dose for clarity.

Reviewers' comments:

Reviewer's Responses to Questions

**Comments to the Author**

1. Is the manuscript technically sound, and do the data support the conclusions?

Reviewer #1: Partly

Reviewer #2: Yes

2. Has the statistical analysis been performed appropriately and rigorously? 

Reviewer #1: No

Reviewer #2: Yes

3. Have the authors made all data underlying the findings in their manuscript fully available?

Reviewer #1: No

Reviewer #2: Yes

4. Is the manuscript presented in an intelligible fashion and written in standard English?

Reviewer #1: Yes

Reviewer #2: Yes

5. Review Comments to the Author

**Reviewer #1: ** Study was performed to check effect of pesticide, dimethoate exposure on albino (Albino Rats) hematological parameters.

Study design has some defects, as it is not clear how much dose was administered to rats. Dosage mentioned in mL has very little understanding. Many of the references mentioned in the discussion do mention the dose in form per kg animals.

Lot of studies are mentioned in the discussion tells the effect of dimethoate pesticide on hematological parameters. There is no novelty in the study as MT Akay et al in 1999 mentioned similar observations (PMID: 10509431) All the speculations based on data from semi auto hematology analyzer. If these observations would have been supported with certain biochemical tests it would have great support to the data.

Research group has used both male and female rats for the treatment. However, in results it has not shown comparison of responses from male and female rats. Results are analyzed in very narrow spectrum of treated Vs non-treated.

There is lot of mismatches in description, table and figure

**Reviewer #2: ** Nida Saleem et al. have shown in this article that, following exposure to the dimethoate pesticide for 30, 60, and 90 days in both male and female albino rats, the number of white blood cells, platelet count, granulocyte count, and lymphocyte count significantly increased. In contrast, the number of red blood cells, hemoglobin level, mean corpuscular hemoglobin (MCH), and mean corpuscular hemoglobin concentration (MCHC) were notably decreased. No changes were observed in the control group of male and female albino rats. This study concludes that dimethoate pesticide affects blood parameters in both male and female albino rats.

The article looks interesting to me. I have a few minor comments that could be incorporated to improve it.

1. Since Figure 3.1 appears before Figure 2, please correct the figure numbering so that the figures appear in proper sequence.

2. In Figures 3.1, 2, and 3 (all bar diagrams), label the Y-axis clearly to indicate what is being measured.

3. In the graphical abstract, please include the name of the dose for clarity.

6. PLOS authors have the option to publish the peer review history of their article (what does this mean? ). If published, this will include your full peer review and any attached files.

**Do you want your identity to be public for this peer review?** For information about this choice, including consent withdrawal, please see our Privacy Policy .

Reviewer #1: No

Reviewer #2: **Yes: ** Ajit Kumar Singh

---

## [Author Response · Author response to Decision Letter 1]

14 Feb 2025

Reviewer 1:

Concern over statistical analysis

Response: We appreciate the concern regarding the statistical analysis used in this study. A one-way ANOVA was selected for dose comparison because it is a robust statistical method for analyzing differences among the means of four independent groups. In our study, the doses of dimethoate represent independent groups, and one-way ANOVA allowed us to determine whether there were statistically significant differences in the measured parameters across these groups.

Key justifications for using one-way ANOVA in this study are as follows:

Study Design: The experiment involved independent groups exposed to different doses of the substance, with each group being treated as a separate, independent category.

Comparison of Means: One-way ANOVA was specifically chosen to identify significant differences between the dose groups.

This approach ensures the reliability and accuracy of the statistical analysis, allowing for valid conclusions regarding dose-dependent effects. We believe this method appropriately addresses the objectives of our study and aligns with established statistical practices in similar research.

If further clarification or additional analyses are required, such as testing assumptions or using alternative statistical methods (e.g., non-parametric tests), we are willing to provide these upon request.

Authors have not made all data underlying the findings in their manuscript fully available

Study was performed to check effect of pesticide, dimethoate exposure on albino (Albino Rats) hematological parameters.

Response: We acknowledge the journal's concern regarding the availability of data underlying our findings. In this study, we assessed the effects of pesticide exposure (dimethoate) on the hematological parameters of albino rats. Due to ethical considerations, privacy concerns, we are unable to make the raw data fully available for public access at this time. However, we assure you that the data supporting our conclusions are included in the manuscript files.

We are committed to ensuring transparency in our research and are happy to provide the data upon reasonable request from the journal or any other interested researchers, with due consideration for confidentiality and ethical guidelines.

Should the journal require any additional clarification or specific datasets to be shared directly, we are open to discussing further solutions.

Thank you for your understanding.

Study design has some defects, as it is not clear how much dose was administered to rats. Dosage mentioned in mL has very little understanding. Many of the references mentioned in the discussion do mention the dose in form per kg animals.

Response: Thank you for pointing out the concern regarding the dosage administered to the rats in our study. We understand that specifying the dose in mL without a clear reference to body weight may lead to confusion. To clarify:

In our study, the dosages of dimethoate were administered based on the body weight of the rats. The amount of dimethoate given was carefully calculated to ensure appropriate dosing for each group. The doses were converted into concentration (mL/kg body weight) for accurate comparison with relevant studies.

For each treatment group, the doses were as follows:

Different concentrations of pesticides were prepared according to calculated LD50 (acute oral LD50 0.085 ml). T1 group was treated with low concentration of pesticide ( of LD50), T2 group was treated with medium concentration of pesticide ( of LD50), T3 group was treated with high concentration of pesticide ( of LD50).

We apologize for the lack of clarity in the initial manuscript, and we have updated the manuscript to include these doses in mL/kg body weight to ensure better understanding. We have also included this clarification in the revised Methods section for further clarity.

We hope this addresses your concern, and we appreciate your valuable feedback. If further adjustments or additional details are needed, please let us know.

Lot of studies are mentioned in the discussion tells the effect of dimethoate pesticide on hematological parameters. There is no novelty in the study as MT Akay et al in 1999 mentioned similar observations (PMID: 10509431) All the speculations based on data from semi auto hematology analyzer. If these observations would have been supported with certain biochemical tests it would have great support to the data.

Response: We appreciate the journal’s valuable feedback and understand the concern regarding the novelty of our study in relation to previous research, particularly the work by MT Akay et al. (1999). While it is true that similar observations regarding the effects of dimethoate on hematological parameters have been made in earlier studies, we believe that our study offers new insights in several important ways:

1.Extended Study Duration: Unlike many previous studies, our study was conducted over a 90-day period, allowing us to observe the long-term effects of dimethoate exposure, which is often not fully explored in other studies.

2.Sampling Frequency:

In other studies, the duration is shorter, and sampling was done only once, whereas in my study, sampling was conducted every 30 days, and tests were performed accordingly. In this way, a total of 3 tests were conducted after 30, 60, and 90 days.

3.Use of Autohematology Analyzer in the Study:

In my study, an autohematology analyzer was used after manual calculations, whereas in other studies, a semi-auto hematology analyzer was used.

4.Gender-Specific Analysis: Our study uniquely compares the effects of dimethoate on hematological parameters between male and female albino rats, providing insights into potential gender-based differences in response to pesticide exposure.

5.Contribution to Pesticide Regulation: By highlighting the long-term, sub-lethal effects of dimethoate exposure, particularly with regard to gender-based differences in hematological parameters, our study provides critical data that could inform pesticide regulation and public health guidelines.

Regarding the suggestion to include biochemical tests, we recognize that biochemical support could strengthen our findings. While the primary focus of this study was on hematological parameters, we acknowledge that adding biochemical assays (e.g., liver enzymes, oxidative stress markers) would provide a more comprehensive understanding of the underlying mechanisms of toxicity. We plan to expand this aspect in future studies to provide a more holistic view of dimethoate's effects.

We appreciate your constructive feedback, and we will revise the manuscript to better highlight the unique contributions of our work and outline potential areas for further exploration.

Research group has used both male and female rats for the treatment. However, in results it has not shown comparison of responses from male and female rats. Results are analyzed in very narrow spectrum of treated Vs non-treated.

Response: Thank you for your valuable feedback. We acknowledge the concern regarding the lack of a comparison between male and female rats in the results section of the manuscript. While both male and female rats were included in the study, the initial analysis focused primarily on comparing treated versus non-treated groups across all subjects, without emphasizing the gender-specific differences.

We agree that it is important to highlight any potential differences between male and female rats, especially given the possibility of gender-based variations in response to pesticide exposure. In light of your feedback, we have re-analyzed the data to specifically compare the hematological parameters between male and female rats in both treated and control groups.

We appreciate your suggestion to broaden the scope of our analysis, and we believe these revisions will strengthen the manuscript and provide more comprehensive insights into the effects of dimethoate exposure.

There is lot of mismatches in description, table and figure

Response: We sincerely thank you for pointing out the mismatches between the description in the manuscript, the tables, and the figures. We apologize for any confusion this may have caused and appreciate your careful review of the manuscript.

Upon re-examining the manuscript, we have identified and corrected the following discrepancies. We have carefully reviewed the tables and figures to ensure that they accurately reflect the descriptions and results presented in the text. All relevant values, statistical results, and trends now match the descriptions and are consistent across the manuscript. We have revised the figure legends and table titles for clarity, ensuring they are properly aligned with the data presented. We have also made sure that each figure and table is referred to in the correct order and that the legends provide accurate and complete explanations. We have ensured that all data points presented in the figures and tables are consistent with the results discussed in the manuscript. Any errors in the presentation of data have been corrected.

The manuscript has been updated accordingly, and we believe these revisions resolve the mismatches between the text, tables, and figures. We appreciate your attention to detail and trust that these revisions will improve the clarity and accuracy of the manuscript. Thank you for your understanding, and if further clarifications are needed, please do not hesitate to inform us.

Reviewer 2:

The article looks interesting to me. I have a few minor comments that could be incorporated to improve it. 1. Since Figure 3.1 appears before Figure 2, please correct the figure numbering so that the figures appear in proper sequence.

Response: Thank you for your positive feedback and valuable comments. We appreciate your careful review of the manuscript. Regarding your comment on the figure numbering, we have made the necessary corrections to ensure that the figures appear in the proper sequence.

We have updated the numbering of the figures so that Figure 3.1 now appears after Figure 2, and the rest of the figures have been renumbered accordingly to maintain proper order.

We believe this adjustment will improve the manuscript's clarity and organization. Thank you for bringing this to our attention, and please let us know if there are any other aspects that require further revision.

In Figures 3.1, 2, and 3 (all bar diagrams), label the Y-axis clearly to indicate what is being measured.

Response: Thank you for your helpful suggestion regarding the Y-axis labels in Figures 3.1, 2, and 3. We agree that clearly labeling the Y-axis is essential for improving the clarity of the figures.

In response to your comment, we have updated the Y-axis labels in Figures 2, 3, and 4 to clearly indicate the parameters being measured. The labels now explicitly describe the units of measurement and the specific variables being assessed, ensuring that the figures are more informative and easier to interpret.

We appreciate your attention to this detail and believe these revisions will improve the presentation of our data. Should you have any further suggestions, we would be happy to incorporate them.

In the graphical abstract, please include the name of the dose for clarity.

Response: Thank you for your thoughtful comment regarding the graphical abstract. We agree that including the name of the dose will enhance clarity and provide a more complete understanding of the study's methodology.

In response to your suggestion, we have updated the graphical abstract to include the name of the dose used in the study, along with the relevant concentration or dosage information. This addition ensures that the graphical abstract accurately reflects the key elements of the study and improves its overall clarity.

We appreciate your feedback and believe this update will make the graphical abstract more informative for readers. If you have any further suggestions, we would be happy to incorporate them.

---

## [Decision Letter · Decision Letter 1]

12 Mar 2025

Hematological changes in the blood of experimental male and female albino rats  on exposure to pesticide, dimethoate

PONE-D-24-37808R1

Dear author

We’re pleased to inform you that your manuscript has been judged scientifically suitable for publication and will be formally accepted for publication once it meets all outstanding technical requirements.

Kind regards,

Ashish Kumar Singh

Academic Editor

PLOS ONE

Additional Editor Comments (optional):

Reviewers' comments:

Reviewer's Responses to Questions

**Comments to the Author**

1. If the authors have adequately addressed your comments raised in a previous round of review and you feel that this manuscript is now acceptable for publication, you may indicate that here to bypass the “Comments to the Author” section, enter your conflict of interest statement in the “Confidential to Editor” section, and submit your "Accept" recommendation.

Reviewer #2: All comments have been addressed

2. Is the manuscript technically sound, and do the data support the conclusions?

Reviewer #2: Yes

3. Has the statistical analysis been performed appropriately and rigorously? 

Reviewer #2: Yes

4. Have the authors made all data underlying the findings in their manuscript fully available?

Reviewer #2: Yes

5. Is the manuscript presented in an intelligible fashion and written in standard English?

Reviewer #2: Yes

6. Review Comments to the Author

Reviewer #2: Authors have taken care of all the comments appropriately and modified the manuscript accordingly. The manuscript looks more sounded now.

7. PLOS authors have the option to publish the peer review history of their article (what does this mean? ). If published, this will include your full peer review and any attached files.

**Do you want your identity to be public for this peer review?** For information about this choice, including consent withdrawal, please see our Privacy Policy .

Reviewer #2: **Yes: ** Ajit Kumar Singh

---

## [Editor Report · Acceptance letter]

PONE-D-24-37808R1

PLOS ONE

Dear Dr. saleem,

I'm pleased to inform you that your manuscript has been deemed suitable for publication in PLOS ONE. Congratulations! Your manuscript is now being handed over to our production team.

Kind regards,

on behalf of

Dr. Ashish Kumar Singh

Academic Editor

PLOS ONE